# Possible Involvement of Long Non-Coding RNAs *GNAS-AS1* and *MIR205HG* in the Modulation of 5-Fluorouracil Chemosensitivity in Colon Cancer Cells through Increased Extracellular Release of Exosomes

**DOI:** 10.3390/ncrna10020025

**Published:** 2024-04-15

**Authors:** Shamin Azwar, Chin Tat Ng, Siti Yazmin Zahari Sham, Heng Fong Seow, Minhian Chai, Mohd Faizal Ghazali, Mohd Faisal Jabar

**Affiliations:** 1Department of Pathology, Faculty of Medicine and Health Sciences, Universiti Putra Malaysia, Serdang 43400, Malaysia; sitiyazmin@upm.edu.my (S.Y.Z.S.);; 2Department of Medicine, Faculty of Medicine, Universiti Kebangsaan Malaysia, Cheras 56000, Malaysia; ngchintat@ppukm.ukm.edu.my; 3School of Animal, Aquatic and Environmental Sciences, Faculty of Bioresources and Food Industry, Universiti Sultan Zainal Abidin, Besut 22200, Malaysia; amthonychai@gmail.com (M.C.); faizalghazali@unisza.edu.my (M.F.G.); 4Department of Surgery, Faculty of Medicine and Health Sciences, Universiti Putra Malaysia, Serdang 43400, Malaysia

**Keywords:** colon cancer, SW480, 5-FU, 5-fluorouracil, chemotherapy resistance, long non-coding RNAs, lncRNAs, mRNAs, GNAS-AS1, MIR105HG, cDNA microarray

## Abstract

A growing number of studies have suggested the involvement of long non-coding RNAs as the key players in not just the initiation and progression of the tumor microenvironment, but also in chemotherapy tolerance. In the present study, generated 5-FU-resistant SW480/DR cells were analyzed via cDNA microarray for its aberrant lncRNAs and mRNAs expression in comparison with the 5-FU-susceptible SW480/DS cells. Among the 126 lncRNAs described, lncRNAs *GNAS-AS1*, *MIR205HG*, and *LOC102723721* have been identified to be significantly upregulated, while lncRNs *lnc-RP11-597K23.2.1-2*, *LOC100507639*, and *CCDC144NL-AS1* have been found to be significantly downregulated. In the meantime, bioinformatic analysis through gene ontology studies of aberrantly expressed mRNAs revealed “regulated exocytosis”, among others, as the biological process most impacted in SW480/DR cells. To investigate, exosome purification was then carried out and its characterization were validated via transmission electron microscopy and nanoparticle tracking analysis. Interestingly, it was determined that the 5-FU-resistant SW480/DR cells secretes significantly higher concentration of extracellular vesicles, particularly, exosomes when compared to the 5-FU-susceptible SW480/DS cells. Based on the lncRNA-mRNA interaction network analysis generated, lncRNA *GNAS-AS1* and *MIR205HG* have been identified to be potentially involved in the incidence of 5-FU resistance in SW480 colon cancer cells through promoting increased release of exosomes into the intercellular matrix. Our study hopes not only to provide insights on the list of involved candidate lncRNAs, but also to elucidate the role exosomes play in the initiation and development of 5-FU chemotherapy resistance in colon cancer cells.

## 1. Introduction

Colon cancer is defined by the progression of abnormal and uncontrolled cellular growth in the inner lining of the colon that originates from non-cancerous polyps. Despite comprehensive and expeditious advancement in surveillance colonoscopy procedures, colon cancer remains as the third-leading cancer burden worldwide involving approximately 1.85 million populations with 883,200 mortality cases reported annually [1]. It is often recognized as a disease of a “Westernized” lifestyle with a temporal trend frequently reported in rapidly developing countries with tendencies of processed meat consumption and a high-fat diet coupled with overall low fiber intake [2,3,4]. The disease is often asymptomatic in patients until it progresses to advanced stages with onset symptoms such as distressing bowel habits, rectal bleeding, abdominal pain, unintended weight loss, and fatigue due to bowel obstruction [5]. For many years, alongside colectomy and lymphadenectomy, colon cancer patients are routinely administered with intravenous infusion of 5-fluoropyrimidine-based adjuvants such as 5-FU (5-fluorouracil). It is administered alongside leucovorin (5-FU/LV) for stage II patients, and together with oxaliplatin (5-FU/LV/oxaliplatin) as a regimen for patients in later stages [6]. For metastatic patients, a combination of 5-FU, oxaliplatin, and irinotecan (FOLFOXIRI) as a rechallenge regimen has been considered, albeit the response rate was only at 38%. In fact, disease relapse remains a major challenge as it is reported in 20–60% of stage II and stage III patients with a respond failure reported in 50–90% of stage IV patients [7,8,9]

Chemotherapy resistance is a common phenomenon across all types of cancer involving increased tolerance of tumor cells against anti-cancer drugs. Despite advances in systemic therapy, the average 5-year survival rate of stage IV patients is still a mere 14.2% while poor chemotherapy response has been observed in 20–50% of stage II and stage III patients where it led to disease progression [7,10,11]. Chemotherapy failure can be attributed to a variety of factors, albeit the development of multi-drug resistance (MDR) still accounts for approximately 90% of cases [9]. At its core, although the mechanism to which MDR is achieved in tumor cells can be multi-faceted and overlapping, it can be briefly summarized into increased drug efflux, decreased drug uptake, increased DNA damage repair, increased resistance to apoptosis, changes in drug metabolism, changes in the drug target, and increased drug compartmentalization [12].

Since their discovery in the early 2000s, long non-coding RNAs (lncRNAs) have gained considerable attention due to their extensive involvement and influence on various intracellular activities, including in the modulation of drug chemosensitivity [13]. By definition, lncRNAs are transcripts that exceed 200 base pairs and are incapable of protein translation due to the absence of an open reading frame (ORF) [14]. In 2006, it was first discovered that lncRNAs, specifically the ectopic expression of *BCAR4* (breast cancer antiestrogen resistance 4), can significantly affect the degree of tamoxifen resistance in ZR-75-1 human breast cancer cells [15]. In the following years, it was reported that the upregulation of *UCA1* expression had been strongly correlated with the incidences of doxorubicin and cisplatin resistance in breast and bladder cancers, respectively, via increased *Wnt6* mRNA expression [16,17]. Downregulation of *GAS5* (growth arrest-specific 5) on the other hand can significantly promote trastuzumab resistance in breast cancer [18]. Additionally, the dysregulation of *HOTTIP* can substantially trigger gemcitabine resistance through the regulation of *HOXA14* mRNA in vitro and in vivo of pancreatic ductal adenocarcinoma [19]. Despite these discoveries, studies that revolve around the involvement of lncRNAs in the development of 5-FU chemodrug in colon cancer cells remain scarce. Hence, in the present study, cDNA microarray and bioinformatic analyses were performed to screen and identify potential lncRNAs that may be involved as well as the biological processes that may be impacted.

## 2. Results

### 2.1. Generation of 5-FU-Resistant SW480/DR Cell Line

5-FU-resistant SW480 human colon adenocarcinoma cells (SW480/DR) kindly donated by Prof. Michael (University of Kent) were made further resistant to 5-FU through continuous exposure to a stepwise increasing concentration of 5-FU treatment while growing the 5-FU-susceptible SW480 human colon adenomacarcinoma cells (SW480/DS) in parallel without any exposure to 5-FU treatment. To evaluate the 5-FU resistance, MTS cytotoxicity assays were conducted on both cells after they were treated with varying concentrations of 5-FU for 72 h to determine their half-maximal inhibitory concentration (IC_50_) values. As shown in Figure 1, the SW480/DR and SW480/DS cells untreated with 5-FU (0 μM) were kept as negative controls.

Treatment with 1 μM of 5-FU concentration can be seen to be sufficient to affect the cellular viabilities of the SW480/DS cells to 92.91%, while the SW480/DS cells remains unaffected with 100.00% cellular viabilities. Significant difference in cellular viabilities (*p* < 0.05) can be seen when treating the cells with 10 μM of 5-FU treatment as SW480/DR survived the treatment better at 92.31% when compared to the SW480/DS cells at 51.76%. Similar patterns of significant cellular viability difference (*p* < 0.05) were also observed at both the 100 μM and 1000 μM 5-FU concentration marks as the SW480/DR cells reported viability percentages of 88.40 and 36.74% versus the 23.06% and 7.35% viabilities reported for SW480/DS cells. Based on these findings, it was collectively demonstrated that the SW480/DR cells exhibited a higher degree of tolerance against 5-FU treatment when compared to the SW480/DS cells with the calculated IC_50_ values of 599.70 μM against 14.97 μM with the calculated resistance index (RI) of 40.06. In other words, the SW480/DR cells are 40.06-fold more resistant to 5-FU treatment than the SW480/DS cells.

### 2.2. Validation of Resistance via Apoptosis Assay

To further validate on 5-FU resistance, apoptosis assays through flow cytometry of both the annexin V-FITC/PI- stained SW480/DR and the SW480/DS cells were conducted upon its exposure to 10 μM and 100 μM of 5-FU concentrations for 72 h. The stained cells were distinguished in four separate cell subpopulations: viable (annexin V^−^ PI^−^), early apoptotic annexin V^+^ PI^−^), late apoptotic (annexin V^+^ PI^+^), and necrotic (annexin V^−^ PI^+^). Data plotted in Figure 2 below were based on the subpopulation of late apoptotic cells.

At 10 μM of 5-FU concentration, a significantly lower apoptotic cell percentage (*p* < 0.05) was reported for the SW480/DR cells at 35.88% against the 42.00% achieved by the SW480/DS cells. The significance in the apoptotic cell subpopulation between the two cells was even greater (*p* < 0.0001) at 100 μM when SW480/DR reported an apoptotic subpopulation percentage of 50.29% when compared to the 64.72% reported for the SW480/DS cells. These findings are in accordance with the MTS cytotoxicity assay previously conducted that had reported a higher percentage of cellular viability of the SW480/DR cells in comparison with the SW480/DS cells when they were both treated with 5-FU chemodrug.

### 2.3. Validation of Resistance via Cell Cycle Distribution Analysis

As shown in Figure 3, cell cycle distribution analyses were performed on both the SW480/DR and SW480/DS cells treated with 100 μM of 5-FU, with cells unexposed to the 5-FU chemodrug (0 μM) serving as the control group. The analysis was conducted to indicate that increased apoptosis observed in the SW480/DS cells post-5-FU treatment was the direct cytotoxic consequence of 5-FU chemotherapy drug. As a TS inhibitor, 5-FU exhibits its anticancer effects through the inhibition of DNA synthesis.

The effects of 5-FU cytotoxicities are evidently lowered on the SW480/DR cells whereby a significant portion of the cells (*p* < 0.05) is still able to enter the S phase of the cell cycle when compared to the SW480/DS cells at 34.8% against 24.0%. The 5-FU-derived cell cycle arrest is evidently more profound in the SW480/DS cells where a large fraction of the populations is arrested at the sub G_0_ and G_0_/G_1_ of the cell cycle phases at 10.3% and 64.0%, respectively, when compared to the 3.3% and 40.0% reported for the SW480/DR cells. At the effects of the 5-FU treatment, SW480/DS failed to enter the G_2_/M phases with the reported population percentage of 1.3%, while SW480/DR remained unaffected at 20.7%.

### 2.4. Differentially Expressed LncRNAs and mRNAs

In this study, purified total RNAs of both the SW480/DR and SW480/DS cells were analyzed using Agilent SurePrint G3 Human Gene Expression v3 8x60k format cDNA microarray and processed in the Agilent GeneSpring GX v14.9.1 software. This was conducted to analyze the genetic expression profiles of the SW480/DR cells in relation to the SW480/DS cells. A total of 58,341 entities based on signal intensities were described and normalized using the “percentile shift normalization” approach with the upper and lower cutoffs of 100% and 20%, respectively, to exclude background noises that were introduced by saturated and bias-heavy probes.

As illustrated in Figure 4, a heatmap was generated based on the hierarchical clustering analysis conducted that effectively grouped together transcripts that share similar expression profiles for better data visualization. Supervised clustering was made by computing specific algorithms in which the Euclidean distance metric and Ward’s linkage rule were met by these transcripts. A volcano plot was then constructed based on fold-change and *p*-values of the gene list. It allowed for a quick visual reference on the relationship between the magnitude of the fold-change and the statistical significance of each gene. Based on the moderated *t*-test (unpaired *t*-test) with Benjamin–Hochberg multiple testing corrections that set a maximum *p*-value of 0.05 and fold-change cutoff of 2.00 for each gene, genes that do not satisfy these criteria were excluded from the end dataset (gray dots). As a result, (C) a total of 340 differentially expressed genes were identified of which 116 and 224 were determined to be upregulated and downregulated transcripts, respectively. (D) In all, 53 transcripts were identified as long non-coding RNAs, 33 transcripts were long intergenic non-coding RNAs (lincRNAs), 214 genes were mRNAs, while the remaining 40 transcripts termed as “misc” were a combination of transcription factors and pseudogenes.

Based on Table 1, it was shown that the top five most significantly (*p* < 0.05) upregulated lncRNAs were *GNAS-AS1*, *MIR205HG*, *LOC102723721*, *LINC00668*, and *lnc-FARS-2* with the fold-changes of 56.86-, 39.69-, 29.62-, 13.28-, and 6.32-folds, respectively. As for mRNAs, *FBP1*, *SMOC*, *CCDC106*, *ARHGEF26*, *and KCTD12* had been demonstrated to be significantly (*p* < 0.05) upregulated with the respective fold-changes of 51.81-, 20.06-, 18.62-, 18.51-, and 18.04-folds. The continuation of this list up to the top 100 upregulated lncRNAs and mRNAs can be found in Appendix A Appendix A.

Based on Table 2, the top five lncRNAs with the lowest downregulation fold-changes are the *lnc-RP11-597K23.2.1-2* (−81.39), *LOC100507639* (−53.46), *CCDC144NL-AS1* (−37.98), *LOC101930053* (33.87), and *lnc-MAOA-2* (−29.26). For mRNA transcripts, *FOXL2* (−161.42), *SPRR2D* (−122.24), *STAT6* (−102.50), *BEST1* (−99.86), and *TRPV6* (−95.98) were best described as the most significantly (*p* < 0.05) downregulated. The continuation of this list up to the top 100 downregulated lncRNAs and mRNAs can also be found in the Appendix A Appendix A.

### 2.5. Validation of Microarray Data via qRT-PCR Technique

To assess on the validity of the lncRNA and mRNA fold-changes data acquired through the cDNA microarray dataset, an in-house RT-qPCR experiment was conducted by using the SW480/DR cells by means of comparison. Out of the many lncRNAs identified in the cDNA microarray experiment, lncRNAs *GNAS-AS1*, *MIR205HG* and *LINC00668* as well as mRNA *FBP1* were selected for the comparison analysis.

As illustrated in Figure 5, no statistically significant fold-change difference (*p* > 0.05) was observed for all of the tested lncRNAs. In *GNAS-AS1*, the in-house RT-qPCR experiment had reported a mean fold-change of 41.5-fold in comparison to the 56.9-fold reported in the cDNA microarray experiment. No statistical significance was also reported between the in-house RT-qPCR and the cDNA microarray experiments with a 46.8-fold against a 39.7-fold determined for lncRNA *MIR205HG* and a 15.4-fold against a 13.3-fold determined for lncRNA *LINC00668*. As for mRNA, the in-house RT-qPCR experiments had yielded a mean fold-change value of 88.95-fold when compared to the 51.82-fold determined via the cDNA microarray data. Although the fold-change difference was distinct at 37.13-fold or nearly 2-fold between each other, no statistical significance was reported (*p* > 0.05). In regard to this matter, it is important to note that the fold-changes acquired by the RT-qPCR method have been proven to be more accurate. All in all, it can be concluded that the fold-changes attained through the qPCR experiments were consistent with the fold-changes obtained through the cDNA microarray analysis, indicating good reliability and reproducibility of the microarray data.

### 2.6. Gene Ontology (GO) Analysis

Based on the list of dysregulated mRNAs identified in the SW480/DR cells in relation to the SW480/DS cells, gene ontology enrichment analysis was performed using Metascape v3.5 software to identify for the most plausible gene function. The mRNA gene list was evaluated for its association with three biological domains, namely in biological processes, molecular functions, and cellular components. With “*Homo sapiens*” as the selected organism, the software then annotates and identifies relevant ontologies based on the GO Resource database (v2021-02) that satisfy the minimum overlap of three; a *p*-value cutoff of <0.01 and a minimum enrichment factor of >1.5. *p*-values were calculated based on the accumulative hypergeometric distribution. The resulting ontologies from each domain are described as “hits”.

A total of eleven hits were identified under the biological processes’ domain with “regulated exocytosis” described as the hit with the greatest significance (Figure 6) with a total of 22 gene counts. Other hits within the same domain include “regulation of NOD2 signaling pathway”, “regulation of wound healing”, “negative regulation of regulated secretory pathway”, “ruffle organization”, “regulation of multi-organism process”, “small GTPase-mediated signal transduction”, “olfactory bulb development”, “positive T-cell selection”, and “positive regulation of type I interferon production”. Meanwhile, under the cellular component domain, only one hit was described, which is the “a band (of sarcomere)”. Similarly, only one hit was identified under the molecular function domain, namely, the “endopeptidase activity”. Within the interest of the studies, the “regulated exocytosis” biological process hit was selected for further scrutiny for its plausible involvement in promoting 5-FU chemotherapy resistance. In growing bodies of the literature, extracellular vesicles, particularly the exosomes, have been described as being involved in enhancing and sustaining drug resistance. To investigate, the conditioned media that were used to maintain both the SW480/DR and SW480/DS cells for 72 h were processed for the purification of its secreted exosomes for downstream analysis.

### 2.7. Characterization of Purified Exosomes

As extracellular vesicles with the size merely of nanometers, exosomes can only be observed and morphologically characterized with the use of transmission electron microscopy (TEM). Morphological observation not only allows for its size estimation but also for the identification of other extracellular vesicles such as microvesicles and apoptotic bodies that may be present in the sample that may skew results. In this study, the exosomal eluates purified from the conditioned media of the SW480/DR cells, hereafter termed Exo/DR, were selected for their morphological viewing. Characterization was also performed via nanoparticle tracking analysis (NTA) that allows for the determination of the particles exact size distribution. In this analysis, both Exo/DR and the exosomal eluates purified from SW480/DS cultures, hereafter termed Exo/DS, were selected. Together, these analyses stand as the gold-standard procedures for confirming the presence of exosomes in a given sample.

When viewed under TEM at 50,000 and 100,000 total magnifications (Figure 7), the observed particles from Exo/DR were approximately the size expected for an exosome at around 50–180 nm. In terms of shape, the particles were morphologically diverse, though the majority appeared to be spherical in shape with a double-layered membrane. Meanwhile, NTA reported a mean particle size of 173.2 ± 6.83 nm and 159.7 ± 11.54 nm for Exo/DR and Exo/DS eluates, respectively. The analysis had also reported a set refractive index and adsorption values of 1.349 and 0.1, respectively. A polydispersity index (PDI) of 0.492 ± 0.069 was determined for Exo/DR while a PDI 0.454 ± 0.083 was determined for Exo/DS eluates. Collectively, both analyses had successfully characterized and confirmed the presence of exosomes in the collected eluates.

### 2.8. Protein Quantification of Purified Exosomes

To confirm the hypothesis that increased “regulated exocytosis” is directly linked to increased activity of exosomal release, a protein quantification assay was conducted on both the eluates of purified Exo/DR and Exo/DS. Protein determination of purified eluate serves as a useful benchmark tool to evaluate the concentration of purified exosome, as exosomes have been well established to express various surface markers on their membranes such as Alix and CD81. As such, the levels of protein concentration are directly proportional to the concentration of purified exosomes.

Interestingly, it was discovered that the Exo/DR eluates that were isolated from 5-FU-resistant SW480/DR cultures contain a significantly higher degree of protein concentrations (*p* < 0.0001) when compared to the Exo/DS eluates isolated from 5-FU-susceptible SW480/DS cultures (Figure 8). Specifically, when 30 mL of conditioned media was tested, a protein concentration of 341.1 μg/mL was reported for the Exo/DR eluates in comparison to the 283.7 μg/mL of protein concentration reported for the Exo/DS eluates. Similarly, when 36 mL of conditioned media was tested, a significantly higher level of protein concentration (*p* < 0.0001) was also determined for the Exo/DR eluates when compared to the the Exo/DS eluates at 399.21 μg/mL against 278.7 μg/mL. All in all, these data serve as evidence to indicate that the 5-FU-resistant cells secrete more exosomes than the 5-FU-susceptible cells and are in accordance with the GO analysis that suggested increased “regulated exocytosis” in the SW480/DR cells when compared to the SW480/DS cells.

### 2.9. LncRNA-mRNA Co-Expression Network Analysis

Based on the identified dysregulated lncRNAs partially described in Table 1 and Table 2, a comprehensive in silico human RNA-RNA predictive interaction software was employed to identify candidate mRNAs that may be potentially regulated by the selected dysregulated lncRNAs. The database, which can be found at http://rtools.cbrc.jp/cgi-bin/RNARNA/index.pl (accessed on 13 September 2022), includes all the predicted lncRNA-mRNA interactions using the 23,898 lncRNA and 81,814 mRNA sequences obtained from the Gencode project. The software evaluates potential lncRNA-mRNA interactions by computing the minimum cellular energy required for such interaction to occur. Following the laws of cellular energy conservation, local and native RNA-RNA interaction would typically benefit from the lowest form of energy usage. By setting the minimum energy threshold to −20 kcal, ten significantly dysregulated lncRNAs based on Table 1 and Table 2 were selected to identify each potential mRNA interaction. These include *GNAS-AS1*, *LINC00237*, *LINC01006*, *PSMG3-AS1*, *LMCD1-AS1*, *MIR205HG*, *LINC00973*, *LINC00668*, *LINC00941*, and *LAMTOR5-AS1*.

The resulting predicted mRNA targets based on the submitted lncRNA list were then cross-checked with the list of dysregulated mRNA genes in the SW480/DR cells from the cDNA microarray dataset whereby mRNAs that are not described to be dysregulated in the dataset were omitted. A lncRNA-mRNA co-expression network was then generated and visualized by using the Cytoscape 3.8.2 software as shown in Figure 9. The full list of the analyzed candidate lncRNA with their predicted mRNA interactions can be found in Appendix A.

In keeping with the study’s desire to investigate further the “regulated exocytosis” hit, the list of mRNAs identified to be interacting with each of the ten lncRNAs in Figure 9 was exported and compared to the list of mRNAs (gene count) identified to be involved in the “regulated exocytosis” process during the gene ontology studies. This is to further narrow down and identify candidate lncRNAs that may be responsible for the regulation of “regulated exocytosis”. As observed in Figure 9B, nine mRNAs (blue dots) were associated with the ontology, which correspond to the eight predicted lncRNA regulators. These include *GNAS-AS1*, *LINC00973*, *MIR205HG*, *PSMG3-AS1*, *LINC00941*, *LMCD1-AS1*, *LINC00668*, and *LAMTOR5-AS1*. Based on this, it is plausible that these lncRNAs may have played a key role in the “regulated exocytosis” biological process in the SW480/DR cells to promote a higher rate of exosome secretion.

## 3. Discussion

Colon cancer has been considered one of the most lethal types of carcinomas, and despite the best efforts to advance surveillance colonoscopy procedures, the 5-year relative survival rate for late-stage presentation among patients with distant metastases is still a mere 13.3% [20]. Chemotherapy failure has been predominantly attributed to the increase in colon tumor cell tolerance against 5-FU-based treatment as the gold-standard chemotherapy regimen for colon cancer patients. While the administration of such regimens as FOXFOXIRI and XELOXIRI (capecitabine + oxaliplatin + irinotecan) can be considered to enhance dose efficacy, the incidences of its adverse events and toxicity profile have been well documented in addition to its nonetheless poor response rate in stage III patients [21,22]. In recent years, the development of chemotherapy resistance has been well established to involve various pathophysiological functions, which also include the abnormal expression levels of lncRNAs and mRNAs [23].

In the present study, a 5-FU-resistant SW480/DR colonic adenocarcinoma cell line model was generated that is significantly much more resistant to 5-FU treatment than the 5-FU-susceptible SW480/DS cell line following the induction methods established by previous works in the literature [24,25,26]. Resistance was thoroughly validated not only through cytotoxicity profiling but also through both apoptosis and cell cycle analyses. Flow cytometry of the annexin V-FITC/PI-stained SW480/DR cells evidently showed lower early (data not provided) and late apoptotic subpopulation when compared to the stained SW480/DS cells upon 5-FU treatment. In their paper, Mhaidat et al. determined that 5-FU-induced apoptosis in SW480 colon cancer cells are caspase-dependent, as it appeared to be initiated by caspase-9 and the activation of the PKCδ [27]. Increased resistance to apoptosis has been known to be one of the signatures of cellular drug resistance [28,29,30]. In HCT29 human colorectal adenocarcinoma cells, increased 5-FU-induced apoptosis has been accounted for by the suppression of PI3K/Akt activation [31]. Similarly, Lin et al. showed that the suppression of the PI3K/AKT pathway leads to the downregulation of *cyclin D1*, *Bcl-2*, and *ABCG2* expression, while it increases the expression of p21 and Bax, which collectively are attributed to the re-sensitization of 5-FU-resistant HCT-8 colon cancer cells [32]. Interestingly, the modulation of PI3K/AKT pathway is not confined to a particular type of chemotherapy agent. It was also highlighted that the pathway can even influence the incidence of MDR through participating in various physiological functions. In breast cancer, the activation of the PI3K/AKT/mTOR pathway has been reported to be associated with increased resistance to chemotherapy [33].

cDNA microarray analysis was then adopted to profile for differential lncRNA and mRNA expression in the 5-FU-resistant SW480/DR cells in comparison with the 5-FU-susceptible SW480/DS cells. Compared to next-generation sequencing (NGS), cDNA microarray was selected in this study due to its low-cost benefit over a larger number of samples and for its convenience as well as familiarity with its procedures for data analysis. Among the 53 differentially expressed lncRNAs, *GNAS-AS1*, *MIR205HG*, and *LOC102723721* ranked as the most upregulated lncRNAs, followed by LINC00668 and *lnc-FAR2-2*, while *lnc-RP11-59K23.2.1-2*, *LOC100507639*, and *CCDC144NL-AS1* were identified as the most downregulated lncRNAs. All 214 aberrantly expressed mRNAs from the cDNA microarray dataset were analyzed in the Metascape (version 3.5, Metascape Foundation) with *Homo sapiens* as the input species to analyze for gene functions through gene ontology studies. Interestingly, the software returned “regulated exocytosis” as the biological process hit with the most gene count and high degree of significance. Based on the database (v2021-02), the term “regulated exocytosis” refers to a process in which soluble proteins and other substances that were initially stored in the secretory vesicles are released through the extracellular environment by docking and fusing themselves with the plasma membrane through a process known as exocytosis. Exocytosis is a transport process commonly used to describe the secretion of exosomes, aside from lysosomal exocytosis. Even so, recent evidence has demonstrated the presence of exosomes inside the lysosome where they are protected from environmental stresses that may lead toward degradation before their release into the extracellular space via lysosomal exocytosis [34]. The role of exosomes in the progression and metastases of colon cancer has been well documented [35]. It has also been shown that colon cancer cells secrete a higher number of exosomes when compared to normal epithelial cells of the gastrointestinal lining [36]. In a report by Augimeri et al., they observed increased production of exosomes in aromatase inhibitor-resistant human breast cancer MCF-7 cells when compared to wild-type MCF-7 cells [37]. In DU145 prostate cancer cells, it was demonstrated that DU145 cells that are resistant to docetaxel secrete 2–3 times higher amounts of exosomes in comparison with docetaxel-sensitive DU145 cells [38]. Increased production of exosomes can also be observed in prostate cancer patients determined resistant to platinum treatment [39].

Hence, to investigate further the “regulated exocytosis” hit, each of the conditioned media used to grow the 5-FU-resistant and 5-FU-susceptible cells were harvested in large volumes for exosomal isolation. For this, the use of a membrane affinity spin column from Qiagen exoEasy Kit was selected instead of the conventional ultracentrifugation method due to its rapid turn-around time and convenient process [40]. Reports have also shown that membrane affinity spin columns not only capture intact vesicles more effectively than the ultracentrifugation method but also introduced lesser background granules that may interfere with morphological observation [41]. Upon isolation, various aspects of the exosomes can be assessed to validate its authenticity and structural integrity. In recent studies, morphological observation through the TEM technique was applied followed by size distribution determination via the NTA method, the combination of which is generally considered as the gold-standard procedure for the validation of exosomal presence [42]. Based on the pictographs, it is observable that these exosomes are diversified in their morphologies and sizes, even though they were from the same cell type and were isolated using similar methods. This is not uncommon. Based on the studies conducted by Zabeo et al., these variabilities can be traced back to the biogenesis of the exosome itself whereby the MVBs may intrinsically harbor structurally diverse vesicles [43]. Interestingly, it was also suggested in their studies that the morphological variability seen is attributed to their different roles and functions. In terms of size distribution, based on a collection of studies, it was concluded that the size range of isolated exosomes should be approximately between 70 to 160 nm in diameter [44]. Interestingly, recent studies have also concluded that the eluates from the 5-FU-resistant cells have higher levels of exosomal protein content than the eluates collected from 5-FU-susceptible cells. This is in accordance with previous studies that showed increased exosome release from drug-resistant cells when compared to parental cells. In studies concerning breast cancer, it was demonstrated that MCF7 and MDA-MB-231 cells resistant to both doxorubicin and paclitaxel are associated with a significant two- to threefold greater exosome release when compared to their parental twin [45]. The approach to quantify exosomes based on the eluates protein concentration via BCA assay is not uncommon. This approach can be found in many other studies that work on exosomes due to its simplicity and short turn-around time [46,47,48].

Identification of candidate lncRNAs that may be involved in promoting increased exosome release was conducted through in silico analysis. The software, termed “rtool”, consisted of a huge database of RNA-RNA interactions that are predicted through the minimum energy calculated. In this study, the rtool software was used to predict lncRNA-mRNA interactions, similar to the works conducted by Yu et al. and Geng et al. [49,50]. Due to our current expanded knowledge on *GNAS-AS1* and *MIR205HG*, these lncRNAs were further explored, and to the best of our knowledge, our study is the first to suggest a direct correlation between *GNAS-AS1* and *MIR205HG* and the incidence of chemotherapy resistance. *GNAS-AS1*, also known as GNAS Antisense RNA 1 as one of the alternative transcripts of human GNAS locus localized on chromosome 20q13.3, has often been disclosed as the lncRNA responsible for tumor initiation and development in various cancers [51,52,53]. In a study conducted by Zhang and colleagues using high-throughput analysis, it was discovered that *GNAS-AS1* is highly upregulated among other lncRNAs in CRC tissues when compared to normal tissues [54]. Meanwhile, *MIR205HG* as the lncRNA that acts as the host gene for *miR-205* has been described as a strong predictor of erlotinib and gefitinib responses in lung cancer cells [55]. It was previously reported that the downregulation of miR-205 may significantly promote cisplatin resistance in prostate cancer cells through the inhibition of autophagic activity [56]. As *MIR205HG* undergoes post-translational processing that leads to the synthesis of *miR-205*, any reported involvement of *miR-205* dysregulation in drug resistance may also reflect on the involvement of *MIR205HG*. The correlation between miR-205 downregulation and drug resistance was also reported in studies concerning docetaxel, doxorubicin, and gemcitabine-resistant cancer cells [57,58]. Together, these reports provide strong indication that lncRNA *GNAS-AS1* and *MIR205HG* may also be involved in the incidence of 5-FU chemotherapy resistance. Interestingly, based on the data presented in this study, it will not be surprising to learn that the mechanism by which this resistance is achieved is through the promotion of increased exosomes release from the resistant cells. Hence, future works may benefit well from designing experiments to assess the exact mechanism and signaling pathways involved, and to confirm the involvement of lncRNA *GNAS-AS1* and *MIR205HG* in sustaining 5-FU resistance in colon cancer cells. Due to the limitation in recent studies, future studies may also benefit from the use of multiple colon cancer cell lines to assess the involvement of these lncRNAs across multiple cancer stages. For better data representation, especially in colon cancer cells, it would be interesting to analyse the expression patterns of these lncRNAs in different Duke’s stages.

## 4. Materials and Methods

### 4.1. Cell Lines and Induction of Drug-Resistant Model

Human SW480 colon adenocarcinoma cell line (SW480/DS) and SW480/DR (5-FU-resistant SW480 colon adenocarcinoma cell line) kindly donated by Prof. Michaelis (University of Kent, UK) were maintained in Iscove’s Modified Dulbecco’s Medium (IMDM), supplemented with 10% fetal bovine serum and penicillin (100 U/mL)/streptomycin (100 mg/mL). The cells were grown as monolayer cultures and were maintained in a humidified atmosphere of 5% CO_2_ at 37 °C. The 5-FU-resistant SW480/DR cells were generated through continuous exposure to increasing concentration of 5-FU for 40 weeks following the methodology reported previously. Only cells in the logarithmic (log) phase of the cell growth were used in all experiments, of which SW480/DR cells were cultured in complete media without 5-FU for 3 days prior to experimentation.

### 4.2. Cell Viability and Proliferation Assay

Both SW480/DR and SW480/DS cell lines were plated into a 96-well plate at a density of 1.5 × 10^3^/well. Following cell attachment after 24 h, the cells were then treated with varying concentrations (1, 10, 100, 1000 and 10,000 µM) of 5-FU for 72 h. Cells that were not exposed to 5-FU treatment (0 µM) served as the control group. Cell viability and proliferation were determined via tetrazolium compound [3-(4,5-dimethylthiazol-2-yl)-5(3-carboxymethophenyl)-2-(4-sulfophenyl)-2H-tetrazolium)] contained in the CellTiter 96^®^ AQ_ueous_ One Solution Cell Proliferation Assay (MTS) following manufacturer’s protocol. Colourimetric evaluations through optical density (O.D.) values were conducted using ChroMate^®^ 4300 (Awareness Technology, Palm City, FL, USA) at 490 nm absorbance wavelength.

### 4.3. Apoptosis Assay

Flow cytometry analysis was conducted following the staining of the cells with FITC-Annexin V Apoptosis Detection Kit (BD Pharmingen, San Diego, CA USA) to measure cellular apoptosis. The cells were plated at 0.3 × 10^3^ cells per each well of a 6-well plate, and after an incubation period of 72 h with 5-FU the cells were digested with 0.25% trypsin and washed twice with ice-cold 1× PBS before they were resuspended in the binding buffer as per manufacturer’s protocol. Analysis was conducted using BD FACSCanto™ (BD Biosciences, Franklin Lakes, NJ, USA) flow cytometer within 1 h following 30 min incubation of the cells at 4 °C with 50 µg/mL of both Annexin V-FITC (fluorescein isothiocyanate) and propidium iodide (PI) dyes.

### 4.4. Cell Cycle Distribution Analysis

Cell cycle phase distributions were determined through cell staining with BD Cycletest™ Plus DNA Kit according to the manufacturer’s protocol. Briefly, harvested cells (seeded at 0.3 × 10^3^/well in a 6-well plate) were trypsinized and washed with 1× PBS before they were resuspended 10 min each in 250 µL of Solution A (trypsin buffer), 200 µL Solution B (trypsin inhibitor), and 200 µL Solution C (20 µg/mL PI dye and RNase A) at room temperature. Analysis was conducted on the BD FACSCanto™ v9.0 (BD Biosciences, Franklin Lakes, NJ, USA) flow cytometer using the FACSDiva™ (BD Biosciences, Franklin Lakes, NJ, USA) software.

### 4.5. RNA Isolation and cDNA Microarray Analysis

Total RNAs from two pairs of SW480/DR and SW480/DS cells were purified and normalized to 50 ng/µL of concentration. A total of 250 ng of RNAs was then reversed transcribed and labeled with cyanine-3 (Cy3) using the One-Color Low Input Quick Amp Labeling Kit (Agilent, Santa Clara, CA, USA) according to the manufacturer’s protocol. Cy3-labeled cRNA was then purified using the RNeasy Mini Kit (Qiagen, Valencia, CA, USA) to purify the amplified cRNA samples before dye incorporation and cRNA yield was assessed via NanoVue™ Plus Spectrophotometer (GE Healthcare, Chicago, IL, USA). After fragmentation steps following the manufacturer’s protocol, the cRNAs were immediately hybridized into the Agilent SurePrint G3 Human Gene Expression v3 8x60K for 17 h at 65 °C in a rotating microarray hybridization oven (Agilent, Santa Clara, CA, USA) prior to washing. Slides were immediately scanned using the Agilent SureScan Microarray Scanner (G4900DA) with 3 µM resolution at 532 nm wavelength. Normalized intensities were extracted using the Agilent Feature Extraction Software v9.1 and were uploaded to the Agilent GeneSpring Software v14.8 for analysis.

### 4.6. Gene Ontology Analysis

Enrichment analyses for dysregulated genes were performed using Metascape v3.5 software at https://metascape.org/gp/index.html#/main/step1) (accessed 13 September 2022) as a powerful annotation analysis tool that integrates several authoritative functional databases such as Gene Ontology (GO), Kyoto Encyclopedia of Genes and Genomes (KEGG), and Uniprot. Gene Ontology (http://geneontology.org/) is a database developed by the Gene Ontology Consortium capable of describing the biological functions of gene sets through three different categories: biological process (BP), cellular component (CC), and molecular function (MF). Meanwhile, the KEGG pathway https://www.genome.jp/kegg/pathway.html is an open database resource that can represent gene sets into known molecular interactions and relation networks. GO analysis allows for the identification of the biological meaning of a large list of dysregulated genes.

### 4.7. Isolation of Exosomes

Purifications of exosomes from serum-free cell culture supernatants were carried out via exoEasy Maxi Kit (Qiagen^®^). In contrast to conventional ultracentrifugation method, the membrane affinity spin columns method allows for rapid isolation time with reported improved in quantity and purity of isolated exosomes. Briefly, the harvested cell culture supernatant was filtered from any cellular fragments and cell debris through 15 min centrifugation at 3000× *g* and at 4 °C. Then, they were mixed with buffer XBP in a 1:1 ratio and were centrifuged at 500× *g* for 1 min via the exoEasy™ spin column, prior to discarding the flow-through. Ten mL of buffer XWP was then added, and the spin columns were centrifuged at 5000× *g* for 5 min to remove any residual buffer, prior to discarding the flow-through. After transferring the spin column to a fresh collection tube, 500 µL of buffer XE was added directly onto the membrane and the spin column was incubated for 1 min, before they were centrifuged at 500× *g* for 5 min to collect the elute.

### 4.8. Transmission Electron Microscopy

For morphological studies, purified exosomes were negatively stained and visualized under transmission electron microscopy (TEM) in the Microscopy Unit, Institute of Bioscience, UPM. Sample preparation involved the eluted exosomes resuspended in cold 1× PBS containing 2% (*w*/*v*) paraformaldehyde (Sigma-Aldrich, St. Lous, MO, USA) as previously described. Then, the exosomes were mounted on copper grids, fixed cold 1× PBS containing 1% (*w*/*v*) glutaraldehyde (Sigma-Aldrich, St. Lous, MO, USA) for 5 min to stabilize the immunoreaction, contrasted by uranyl-oxalate solution for 5 min at pH 7, and embedded by methyl-cellulose-UA for 10 min on ice. Viewing was conducted through LEO 912AB TEM (Carl Zeiss, Oberkochen, Germany) with Omega energy filter, and the images were captured and using Proscan camera controlled by the EsivisionPro v3.2 software.

### 4.9. Nanoparticle Tracking Analysis

Precise assessment of exosome size distribution and concentrations was evaluated through nanoparticle tracking analysis. Briefly, the eluate samples were diluted further in buffer XE and gently agitated to ensure a complete homogeneity. The samples were then loaded into a clean 1 mL glass cuvette before being placed inside the NanoSight LM14 (Malvern Panalytical, Wocestershire, UK) for analysis. A refractive index of 1.349 was selected based on the buffer XE used with the absorption value of 0.1. Raw data containing the values of X intensities were first analyzed in Microsoft Excel prior further statistical analysis in GraphPad Prism v9 software.

### 4.10. Protein Quantification

Purified exosomes were assessed for their protein concentrations via Pierce™ BCA Protein Assay kit (Thermo Scientific, Waltham, MA, USA) according to manufacturer’s protocol with a nine-point calibration standard points ranging from 0 to 2000 µg/mL. Briefly, 25 µL of each standard and eluate samples were mixed with 200 µL of prepared working reagent (WR) solution. The microplates were incubated at 37 °C for 30 min before the absorbance of each standard and eluate sample well was measured at 562 nm wavelength. The absorbance readings of each well were subtracted with the absorbance readings of the set blanks before a standard curve was generated using the nine-point calibration standards. Protein concentrations of each eluate sample (in triplicates) were determined through extrapolating their absorbance readings from the generated standard curves.

### 4.11. LncRNA-mRNA Interaction Network Analysis

LncRNA-mRNA interaction networks were constructed by comparing the bioinformatic analyses from various lncRNA and mRNA databases. First, the appropriate number of lncRNAs was shortlisted and selected to participate based on their significant dysregulation in the dataset. Each lncRNA was then analyzed using the “rtool” software to predict targeted mRNAs based on the minimum energy required for their interaction. A strong interaction is represented by a lower minimum energy value. Then, this lncRNA-mRNA gene list was then compared with the list of mRNAs involved in a given pathway provided by the Metascape v3.5 software.

### 4.12. Statistical Analyses

Statistical analyses were conducted using GraphPad Prism v7.0 software with the appropriate one-way ANOVA and/or Student’s *t*-test employed depending on the experimental setup. All assays were conducted in triplicates (*n* = 3) with at least three independent experiments before their mean standard deviation (SD) was calculated. Experimental data with *p*-value lower than 0.05 (*p* < 0.05) were considered as statistically significant.

## Figures and Tables

**Figure 1 ncrna-10-00025-f001:**
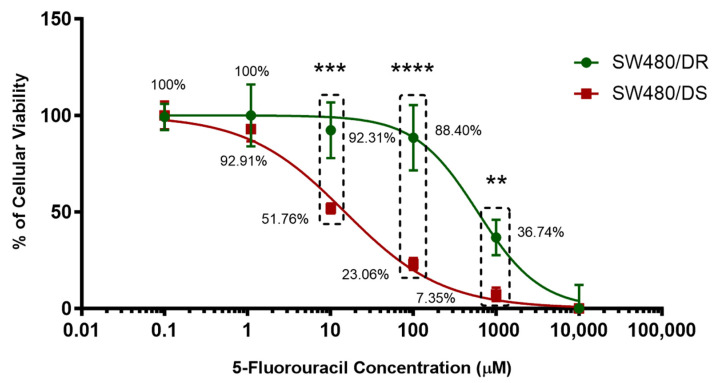
Establishment of 5-FU-resistant SW480/DR colon adenocarcinoma cell line. Upon treatment with varying concentrations of 5-FU chemodrug, the cellular viability percentages of both SW480/DR and SW480/DS were determined through MTS cytotoxicity assay. Data shown are from three independent experiments (in triplicates) with calculated SD values to represent error bars. Significance in cellular viability differences is represented by ** (*p* < 0.05), *** (*p* < 0.001), and **** (*p* < 0.0001).

**Figure 2 ncrna-10-00025-f002:**
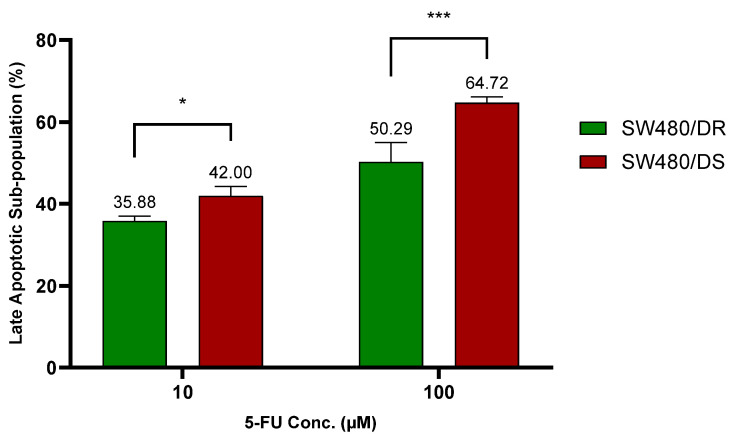
Lower percentages of late apoptotic subpopulation were determined in SW480/DR cells in relation to SW480/DS cells when they were treated with 10 μM and 100 μM of 5-FU concentrations. Analysis of apoptotic subpopulations was conducted by a flow cytometer instrument using cells that were stained with annexin V-FITC/PI dyes. Data shown are from three independent experiments (in triplicates) with calculated SD values to represent error bars. Significance in cellular viability differences is represented by * (*p* < 0.05) and *** (*p* < 0.001).

**Figure 3 ncrna-10-00025-f003:**
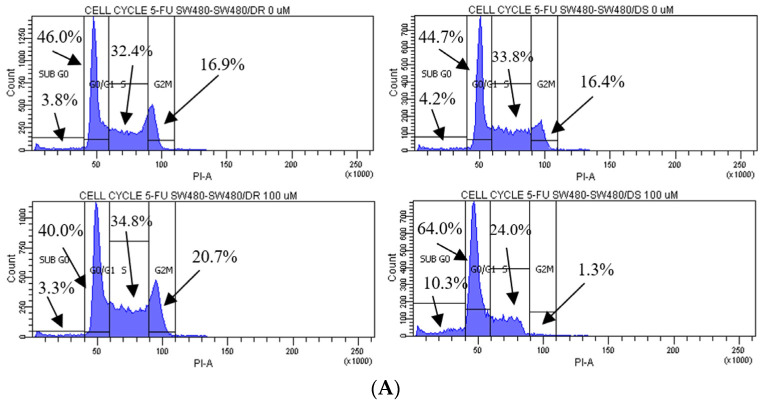
(**A**) Flow cytometric data representation of annexin V- and PI-stained SW480/DR and SW480/DS cells when exposed to 0 μM and 100 μM of 5-FU concentrations using a BD FACSCanto II flow cytometer. (**B**) Percentages of cell cycle phases were divided into four categories, namely the Sub G_0_, G_0_/G_1_, S, and G_2_/M phases. Data shown are from three independent experiments (in triplicates) with calculated SD values to represent error bars. Significance in subpopulation percentages is represented by * (*p* < 0.05), *** (*p* < 0.001) and **** (*p <* 0.0001).

**Figure 4 ncrna-10-00025-f004:**
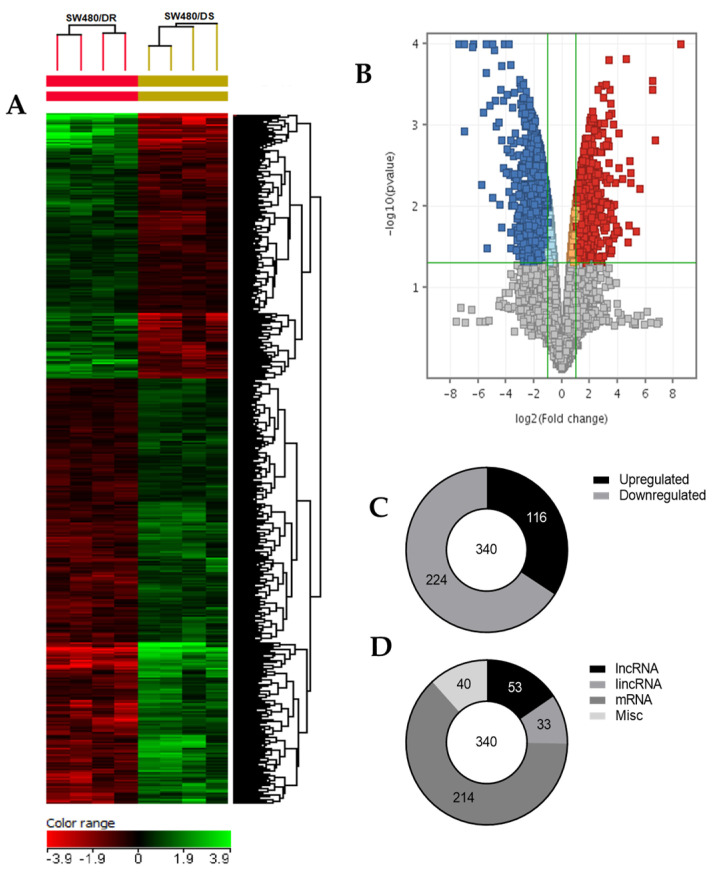
(**A**) Two-dimensional supervised hierarchical clustering analysis of gene expression profiles. Each vertical axis represents a cell line of either SW480/DR or SW480/DS (color bars; red = upregulated, green = downregulated, black = no change). (**B**) Volcano plot generated using GeneSpring GX v14.9.1 software with moderated *t*-test and Benjamin–Hochberg testing correction that depicts the distribution of upregulated (red) and downregulated (blue) gene expression fold-changes (log_2_) and corrected *p*-values (−log_10_). (**C**) Distribution of upregulated and downregulated genes, as well as (**D**) the types of transcripts described upon comparing samples of SW480/DR against SW480/DS. Data shown are from four independent experiments (in one replicate) to achieve the corresponding *p*-values.

**Figure 5 ncrna-10-00025-f005:**
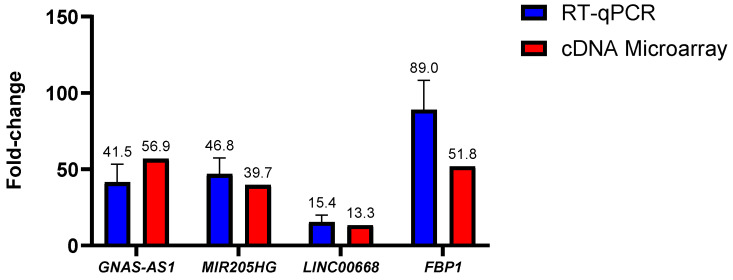
Difference in lncRNA and mRNA fold-changes between the cDNA microarray and in-house RT-qPCR experiments. Data shown are from three independent experiments (in triplicates) with calculated SD values to represent error bars and their statistical analyses were conducted using one-way ANOVA.

**Figure 6 ncrna-10-00025-f006:**
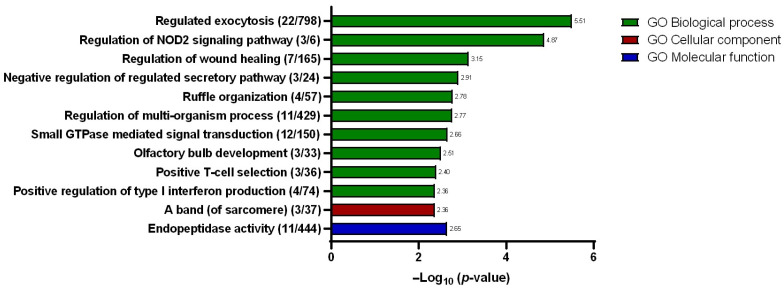
Resulting gene ontologies that are associated with the gene list submitted to Metascape v3.5 software were divided into three different domains: biological processes, cellular components, and molecular functions. Gene counts are stated in brackets “()”, while the degree of significance (*p*-value) is represented in −log base 10.

**Figure 7 ncrna-10-00025-f007:**
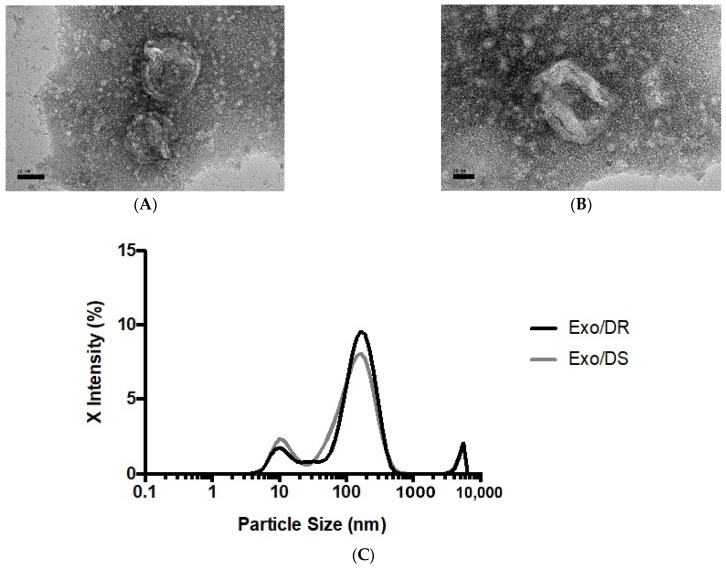
Representative transmission electron photomicrograph at (**A**) 50,000× and (**B**) 100,000× total magnifications of isolated exosomes deriving from SW480/DR cells. The sizes of the exosomes were within the acceptable range from 30 to 180 nm (scale bars; (**A**) 50 nm, (**B**) 20 nm). (**C**) The sizes of the purified exosomes from both SW480/DR (Exo/DR) and SW480/DS (Exo/DS) cultures were confirmed via nanoparticle tracking analysis.

**Figure 8 ncrna-10-00025-f008:**
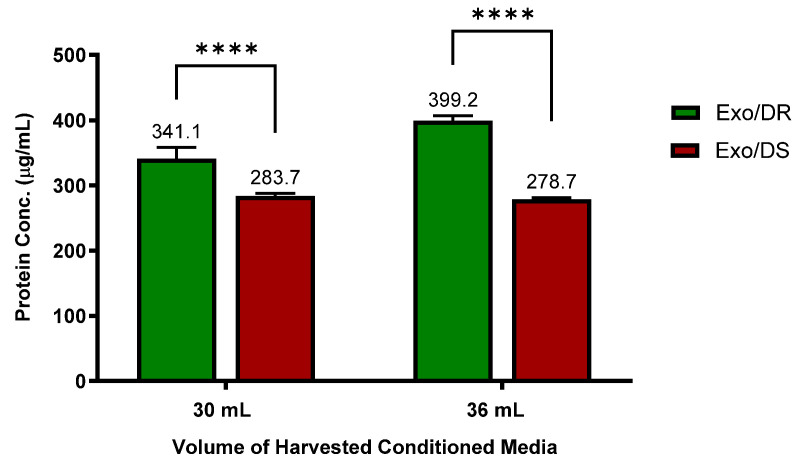
Higher concentrations of exosomes were determined in Exo/DR eluates when compared to exosomal eluates isolated from SW480/DS cultures (Exo/DS). Exosomes were purified using Qiagen exoEasy Maxi Kit according to manufacturer’s protocol before their protein concentration were quantitated using Thermo Scientific Pierce™ BCA Protein Assay Kit. Data shown are from three independent experiments (in triplicates) with calculated SD values to represent error bars and their statistical analyses were conducted using two-way ANOVA and Sidak’s multiple comparisons test. Significance in protein concentrations is represented by by **** (*p* < 0.0001).

**Figure 9 ncrna-10-00025-f009:**
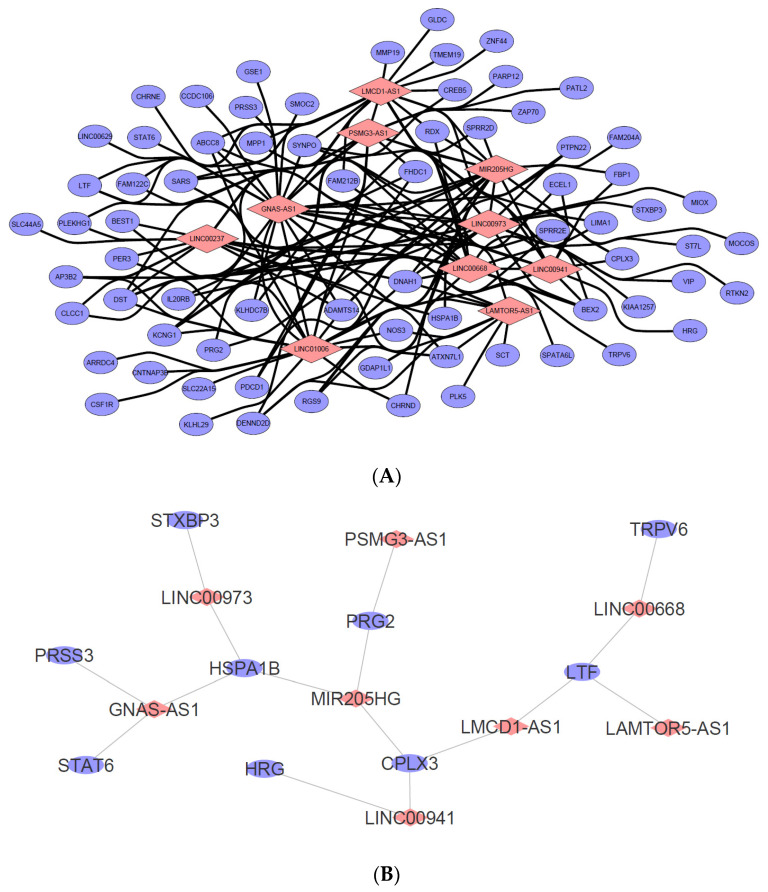
(**A**) LncRNA-mRNA interaction network was constructed based on the ten selected candidate lncRNAs from the cDNA microarray dataset. The resulting mRNA interactions were predicted using the “rtool” database with −20 kcal as the minimum energy threshold and were filtered to show only mRNAs that were also dysregulated in the dataset. Data were visualized using Cytoscape v3.8.2. (Pink diamond = candidate lncRNA, blue circle = predicted putative mRNA targets). (**B**) A total of nine mRNAs were identified as both responsible in the “regulated exocytosis” hit as well as highly likely to be regulated by the candidate lncRNAs listed in (**A**). Data were visualized using Cytoscape v3.8.2. (pink diamond = candidate regulator lncRNA, blue circle = mRNA predicted involved in the biological process).

**Table 1 ncrna-10-00025-t001:** Top ten upregulated lncRNAs and mRNAs and their respective fold-changes and *p*-values based on the cDNA microarray dataset. The transcripts were statistically analyzed in the Agilent GeneSpring GX (v14.9.1) software using moderated *t*-test and Benjamin–Hochberg multiple testing corrections and had satisfied the *p*-value of <0.05 and fold-change cutoff of ±2.00. Data shown are from four independent experiments.

Upregulated lncRNA	Fold-Change	*p*-Value	Upregulated mRNAs	Fold-Change	*p*-Value
*GNAS-AS1*	58.86	0.0456	*FBP1*	51.81	0.0491
*MIR205HG*	39.69	0.0491	*SMOC2*	20.06	0.0456
*LOC102723721*	29.62	0.0477	*CCDC106*	18.62	0.0456
*LINC00668*	13.28	0.0456	*ARHGEF26*	18.51	0.0456
*lnc-FARS2-2*	6.32	0.0456	*KCTD12*	18.04	0.0491
*LOC100506379*	5.39	0.0491	*C11orf39*	17.18	0.0456
*LOC100507002*	4.92	0.0491	*GAGE7*	16.43	0.0491
*LINC01431*	4.49	0.0491	*PER3*	13.85	0.0456
*LOC255187*	4.34	0.0491	*ZNF44*	12.52	0.0456
*lnc-HORMAD2-1*	4.36	0.0456	*CNTNAP3B*	12.22	0.0456

**Table 2 ncrna-10-00025-t002:** Top ten downregulated lncRNAs and mRNAs and their respective fold-changes and *p*-values based on the cDNA microarray dataset. The transcripts were statistically analyzed in the Agilent GeneSpring GX (v14.9.1) software using moderated *t*-test and Benjamin–Hochberg multiple testing corrections and had satisfied the *p*-value of <0.05 and fold-change cutoff of ±2.00. Data shown are from four independent experiments.

Upregulated lncRNA	Fold-Change	*p*-Value	Upregulated mRNAs	Fold-Change	*p*-Value
*Lnc-RP11-597K23.2.1-2*	−81.39	0.0491	*FOXL2*	−161.42	0.0456
*LOC100507539*	−53.46	0.0456	*SPRR2D*	−122.24	0.0491
*CCDC144NL-AS1*	−37.98	0.0491	*STAT6*	−102.50	0.0491
*LOC101930053*	−33.87	0.0456	*BEST1*	−99.86	0.0456
*Lnc-MAOA-2*	−29.26	0.0491	*TRPV6*	−95.98	0.0456
*LOC101929174*	−25.96	0.0491	*SPRR2C*	−90.04	0.0491
*LINC00237*	−20.20	0.0456	*SPRR2E*	−88.79	0.0491
*LOC101930048*	−19.80	0.0456	*SLIT1*	−82.73	0.0456
*LAMTOR5-AS1*	−19.25	0.0456	*GSE1*	−77.99	0.0456
*Lnc-HSPB7-1*	−16.55	0.0456	*CREB5*	−64.42	0.0491

## Data Availability

Not applicable.

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
