# Peer review of "Possible Involvement of Long Non-Coding RNAs GNAS-AS1 and MIR205HG in the Modulation of 5-Fluorouracil Chemosensitivity in Colon Cancer Cells through Increased Extracellular Release of Exosomes"

_ncrna, 2024, doi:10.3390/ncrna10020025_

Round 1

Reviewer 1 Report

Comments and Suggestions for Authors

Azwar and colleagues presented an interesting research article aimed at elucidating the role of non-coding RNA in the development of 5-FU drug resistance in colorectal cancer. For this purpose, the authors used a single CRC cell line sensitive and resistant to 5-FU by performing microarray and RT-qPCR investigations. The results obtained were computationally analyzed to evaluate the pathway responsible for drug resistance as well as the molecular determinants of such resistance, including lncRNA. Overall, the research idea is interesting but the methodologies are weak and based on the use of a single cell line (major concern). Below are reported minor/major comments that will improve the manuscript: 

1) I suggest to change the title by removing “bioinformatic analyses” as the study is based on in vitro experiments;

2) The entire manuscript needs English editing performed by an English native speaker;

3) In the following paragraph, please provide also more updated references “For many years, alongside colectomy and lymphadenectomy, colon cancer patients are typically administered with intravenous infusion of 5-fluoropyrimidine-based adjuvants such as 5-FU plus leucovorin (5-FU/LV) for stage II, or FOLFOX (5- FU/LV plus oxaliplatin) for stage III and stage IV as standard regimens [6]. However, disease relapse still occurs in 40-60% of stage II and III patients, while 50-90% of stage IV patients had reported a failure to respond to the 5-FU regimen [7], [8].”. For this purpose, please see:

- https://doi.org/10.1016/j.cell.2023.02.038

- https://doi.org/10.3390/cancers12092679

- https://doi.org/10.1007/s12029-017-0001-3

- https://doi.org/10.1186/s12885-022-09889-3

3) Improve the grammar of the following sentence: “Wild-type or drug-susceptible SW480/DS cells and 5-FU-resistant SW480/DR cells were further exposed to the consistent but gradual exposure of 5-FU concentration in culture to further strengthen its resistance towards 5-FU treatment.”;

4) The data reported in Figure 1 are not convincing. The graph showed linear regression that could bias your findings. You have to re-analyze your MTS data by using non-linear regression and correctly establishing the killing doses. Please address this issue;

5) In the Methods section, please add a paragraph on “Statistical analyses”;

6) Check the grammar of: “To  validate,  an  apoptosis  assay  was  also  conducted  on  both  SW480/DR  and SW480/DS cells treated with varying concentrations of 5-FU ranging from 10, 100, and 1000 μM.”; 

7) In Figure 2, how do you explain the higher apoptotic rates observed at lower doses of 5-FU in resistant cells? It would be expected higher apoptotic rates at very high concentrations of 1000 uM as observed for 5-fu-sensitive cells. Please argue this issue;

8) The data reported in Figure 3 are not concordant with those reported in Figure 2. In particular, Figure 2 shows 50.29% of apoptotic cells, while in the flow cytometry analysis of Figure 3 the percentages are totally different. Please clarify this critical issue;

9) The data reported in Figure 6 are not convincing. E.g FBP1 levels were almost 2-fold higher when analyzed with RT-qPCR compared to the data observed with cDNA microarray. Please, clarify these data that seem not “relatively comparable” as stated by the authors;

10) In the results section, please do not use “p=0.0000” but “p<0.0001”;

11) In the following sentence of the Discussion section please briefly mention irinotecan and oxaliplatin as further chemotherapy treatments for CRC and provide references: “Chemotherapy failure has been predominantly attributed to the increase in colon tumour cell tolerance against 5-FU-based treatment as the gold-standard chemotherapy regimen for colon cancer patients.”. For this purpose, please see:

- https://doi.org/10.1016/j.cell.2023.02.038

- https://doi.org/10.1007/s12029-017-0001-3

- https://doi.org/10.1186/s12885-022-09889-3

- https://doi.org/10.1186/s12885-021-07823-7

12) Please check the following paragraph: “Microarray analysis was adopted to profile for differential lncRNA and mRNA expression in the 5-FU-resistant cell line in comparison with the parental 5-FU-susceptible cell line. Unlike next-generation sequencing. In comparison with next-generation sequencing (NGS), DNA microarray was selected in this study due to its low-cost benefit over a larger number of samples and for its convenience as well as familiarity with its procedures for data analysis.”; 

13) A limiting aspect of the study is the use of a single CRC cell line. It would be interesting to evaluate differences in lncRNA expression in other CRC cell lines harboring different oncogene mutations. Please argue this critical issue;

14) In the following sentence of the method section, please indicate the doses used for the treatment of cells: “Following cell attachment after 24 hours, the cells were then treated with varying concentrations of 5-FU for 72 hours.”.

Comments on the Quality of English Language

The manuscript needs English editing

Author Response

Dear Reviewer 1,

Kindly please find the attachment attached for the response to your comments.

Reviewer 2 Report

Comments and Suggestions for Authors

The authors of the manuscript entitled "Bioinformatic analyses revealed long non-coding RNAs GNAS-AS1 and MIR205HG as potentially involved in the modulation of 5-Fluorouracil chemosensitivity in colon cancer cells" have presented their data with the following findings. The authors have shown in their manuscript though the generation of resistant colon cancer cells SW480 to 5-Fluorouracil and through bioinformatic analysis that certain lncRNAs such as GNAS-AS1 and MIR205HG which could be involved in the modulation of 5-Fluorouracil chemosensitivity in colon cancer cells. They also carried out a lncRNA-mRNA interaction analysis and identified that that nearly 34 lncRNAs could potentially be involved in the regulation of some biological processes as well as the resistance of colon cancer cells to 5-FU. While the authors have validated the resistance via apoptosis as well as cell cycle distribution and used these cell lines to draw a differential gene expression of lncRNAs and mRNAs using a cDNA microarray with a list of both upregulated and downregulated lncRNAs and mRNAs to drive home their point.

The manuscript is well written however, I feel the authors should address the following questions to support their claims.

1)    The authors carry out experimental data for the first half of the manuscript and suddenly shift to bioinformatic analysis which is purely based on the merit of the bioinformatic analysis. I strongly believe that the authors should use the bioinformatic data to narrow down on their hits and select a few of their hits to further drive home their point.

2)    The authors in figure 6 show the fold-changes of both lncRNA GNAS-AS1 and mRNA FBP1 achieved through 307 in-house RT-qPCR. It would be best to express their data as the levels of lncRNAs or mRNA from their RTqPCR data in control and resistant cells. This should be carried out with atleast the top 5 or few hits hits so as to show the direction of the research hypothesis.

3)    The authors then show lncRNA and mRNA interaction analysis to show candidate and predictive mRNA targets and then simmer down to 9 mRNA predicted to be associated with the GO term regulated exocytosis and one mRNA assocated with AMPK signalling pathway.And thereafter abruptly conclude the manuscript.

4)    I strongly believe that the authors have a strong resource at their disposal, and they should make maximum use of their generated resistance lines and carry further analysis in this direction.

5)    To start with their strategy to create the resistant cell line by gradual increases in drug concentration can help to throw light on the mechanism of resistance. The authors can refer to many manuscripts that focus on how such resistance is created by looking closely at well-known signaling pathways axis such as PI3K-AKT-MTOR and other such pathways. They can substantiate their data with protein levels through western blots.

6)    Finally, the authors can also substantiate their findings by gain of function and loss of function studies of lncRNAs or the mRNAs and follow up with cell viability assays. They can also show the loss of association between the lncRNAs and the associated mRNA and see if their hypothesis holds true.

Overall, I believe the authors should carry out some additional experiments to drive home their point rather than depend on bioinformatic analysis to carry their story.

Author Response

Dear Reviewer 2,

Kindly please find the attachment for the response to your comments.

Round 2

Reviewer 1 Report

Comments and Suggestions for Authors

Dear Authors,

the revised version of the manuscript appears now more detailed and complete. You well addressed all of my comments and the manuscript benefits from your revisions. I have no further comments.

Reviewer 2 Report

Comments and Suggestions for Authors

I thank the authors of the manuscript for incorporating the suggestions in their manuscript. Additionally, the manuscript appears wholesome in its story line and in the research direction.